# Advancements in Triboelectric Nanogenerators (TENGs) for Intelligent Transportation Infrastructure: Enhancing Bridges, Highways, and Tunnels

**DOI:** 10.3390/s23146634

**Published:** 2023-07-24

**Authors:** Arash Rayegani, Ali Matin Nazar, Maria Rashidi

**Affiliations:** 1Centre for Infrastructure Engineering, Western Sydney University, Kingswood, NSW 2747, Australia; a.rayegani@westernsydney.edu.au; 2Zhejiang University/University of Illinois at Urbana-Champaign Institute, Zhejiang University, Haining 314400, China; matin.22@intl.zju.edu.cn

**Keywords:** self-powered sensors, energy harvesting, intelligent road environments, triboelectric nanogenerators (TENG), structural health monitoring

## Abstract

The development of triboelectric nanogenerators (TENGs) over time has resulted in considerable improvements to the efficiency, effectiveness, and sensitivity of self-powered sensing. Triboelectric nanogenerators have low restriction and high sensitivity while also having high efficiency. The vast majority of previous research has found that accidents on the road can be attributed to road conditions. For instance, extreme weather conditions, such as heavy winds or rain, can reduce the safety of the roads, while excessive temperatures might make it unpleasant to be behind the wheel. Air pollution also has a negative impact on visibility while driving. As a result, sensing road surroundings is the most important technical system that is used to evaluate a vehicle and make decisions. This paper discusses both monitoring driving behavior and self-powered sensors influenced by triboelectric nanogenerators (TENGs). It also considers energy harvesting and sustainability in smart road environments such as bridges, tunnels, and highways. Furthermore, the information gathered in this study can help readers enhance their knowledge concerning the advantages of employing these technologies for innovative uses of their powers.

## 1. Introduction

Triboelectric nanogenerator (TENG) technology was first presented in 2012, and since then, it has become one of the most significant technological advances in the world of energy collection [1,2,3]. The TENG has had a great deal of success up to this point, especially in the fields of energy harvesting and self-powered sensors [4]. It begins with the basic operating principles of the triboelectric effect and electrostatic induction, but it may convert nearly every sort of mechanical energy that we encounter in our daily lives into electricity. There has been great success in optimizing the output performance of TENGs [5,6,7]. TENGs can also be utilized as active sensors that are capable of detecting a wide range of environmental factors while operating independently from an external power source [8,9,10,11,12,13,14]. It can be employed in smart control systems within structural systems, facilitating the provision of indispensable power in the absence of external power during catastrophic occurrences [15,16]. Triboelectric and electrostatic induction processes enable the TENG to convert a wide variety of the mechanical energy existing in its surrounding environment into electricity [17,18]. TENGs have been used in a variety of industries as a potential energy harvesting technology due to their multiple advantages, including a high level of efficiency [19,20], constructability [21], and robustness and reliability [22,23,24,25,26,27,28,29,30]. TENGs are a better fit for the low-frequency vibrations in subway systems than other technologies like piezoelectric harvesters or electromagnetic harvesters. Numerous studies have been carried out on TENG-based energy harvesters for transportation infrastructures such as bridges, highways, and tunnels [31,32,33,34,35,36,37,38,39,40,41,42,43,44,45]. The main focus has been on the utilization of innovative materials, production, stability, structural design, and constructability. TENGs have unique qualities and uses because of their high power density, efficiency, and ability to adapt to different situations [46,47,48,49,50,51,52,53,54]. Considering the fact that road quality has a direct impact on the safety of the drivers [55,56,57,58,59,60], intelligent roads can significantly improve the safety and convenience of drivers [52,61,62,63,64,65,66,67,68]. In addition, more complex construction is needed for many specialty roads, including tunnels and bridges. In this regard, nanogenerators can significantly advance the field of real-time sensing and reduce the cost of upcoming smart advancements [69]. In addition, the potential materials are crucial for optimizing the performance and efficiency of TENG-based sensors in each specific application. For bridges, polymeric materials (PDMS, PTFE, PVDF) and conductive materials (ITO, graphene, CNTs) are commonly used, along with flexible substrates and protective encapsulation materials. Tunnels require materials with high durability and resistance to harsh conditions. High robustness, flexibility, and weather resistance are key considerations for materials used in highway applications. Smart infrastructure for energy harvesting relies on materials with high piezoelectric or triboelectric properties. Self-powered vehicle sensors employ flexible polymers, conductive materials, and pressure-sensitive elements. Smart pedestrian crossing systems require materials with good triboelectric properties and mechanical resilience for their triboelectric layers and structural components, respectively. Overall, the selection of appropriate materials plays a vital role in achieving optimal performance in TENG-based, self-powered sensors for transportation applications. The purpose of this study is to provide an introduction to TENGs for smart roads in order to clarify the current state of knowledge regarding triboelectric nanogenerators. The fundamental physical modes of triboelectric nanogenerators are addressed in Section 2. The third section includes a quick overview of self-powered sensors for use in intelligent road environments. Section 4 discusses the challenges, viewpoints, and thoughts related to the creation of self-powered sensors based on TENGs for use in bridges, tunnels, and roads. Therefore, the information in this article can help identify future research avenues in the fields of sensing, sustainability, and energy harvesting.

## 2. Fundamental Physics Modes of Triboelectric Nanogenerators

Triboelectric nanogenerators, more often known as TENGs, are an innovative type of energy generator that was publicly shown for the first time in 2012 [69,70,71,72,73,74,75,76,77]. This electricity can be utilized in a range of different applications, such as energy harvesting and self-powered sensors for transportation engineering. There are four basic physical modes that a TENG can work in, which are shown in Figure 1. These modes are contact separation, lateral sliding, single-electrode, and free-standing. Electrostatic induction charges are all transported to the electrodes by means of these various TENG modes. Figure 1 illustrates how two electrodes are required for all TENG settings. The only exception to this rule is the single-electrode mode [78]. 

A TENG generates alternating current as a result of its reciprocating action. The contact separation mode of a TENG is depicted in Figure 1a. Two electrodes are used in the contact separation mode behind the TENG layers, and a multimeter is attached between the two electrodes to measure the output voltage. Figure 1b depicts a TENG in the lateral sliding mode. The lateral sliding mode has two sensors below the TENG sections, just like the contact separation mode. The relative movement of TENG layers causes a non-electrostatic situation, which is followed by a potential difference. The voltage generated by the potential difference may be measured by utilizing the TENG layer’s reciprocating movement and a voltmeter connected to two electrodes. Figure 1c shows a TENG in single-electrode mode. One method of using a single electrode coupled to an external load is to provide a single-electrode mode. The free-standing state of a TENG is depicted in Figure 1d, in which the TENG layer travels independently of the electrode and slides across electrodes, causing a potential difference. Figure 1e illustrates the equivalent circuit model [78].

## 3. Overview of Self-Powered Sensors for Intelligent Road Environments

Many sensing technologies have been created due to the rapid growth of structural health monitoring systems [80,81,82,83,84,85,86,87,88,89]. Monitoring the structural health (SHM) of transportation infrastructure is crucial not just for reducing economic losses from traffic delays and diversions but also for averting catastrophic collapses and loss of life. In recent years, wireless sensor technologies have been widely implemented in the development of SHM platforms for roads, bridges, and tunnels. Batteries have a limited lifespan and are expensive to replace, which makes wireless sensor systems prohibitively expensive in many situations. Energy harvesting is a viable method for addressing this issue [90,91,92,93,94,95,96,97,98]. Researchers proposed a novel wireless sensor system that gathers vibrations from passing vehicles and converts them into usable electrical energy using TENG [99]. 

Table 1 provides a comprehensive summary of the various methods of vehicle energy harvesting. It highlights the following approaches: kinetic energy harvesting, which captures the vehicle’s kinetic energy during braking or through friction; vibration energy harvesting, which converts vehicle vibrations into electrical energy using piezoelectric nanogenerators; solar energy harvesting, which utilizes solar panels to harness sunlight for electricity; thermoelectric energy harvesting, which capitalizes on temperature differences to generate electrical energy; wind energy harvesting, which captures the kinetic energy from airflow using wind turbines; electromagnetic induction, which harvests energy from mechanical movement or magnetic fields; triboelectric nanogenerators, which convert mechanical energy from motion or vibrations using the triboelectric effect; and piezoelectric nanogenerators, which produce electricity in response to mechanical strain or vibrations. Table 1 serves as a valuable resource for understanding the diverse approaches to vehicle energy harvesting. In this research, self-powered sensors for bridge, tunnel, and road devices based on TENGs are described. In addition, a quick overview of intelligent infrastructure to collect energy from roadways, intelligent pedestrian crossing systems, and monitoring driving behaviors are provided.

### 3.1. Self-Powered Sensor Based on the Triboelectric Nanogenerators for Bridges

Triboelectric nanogenerators can be the source of power for the health monitoring of bridges (please see Figure 2) [25,100,101,102,103,104,105,106]. The newly developed magnetically circular layers of TENGs (MCL-TENG) for velocity sensing and damage detection are illustrated in Figure 2a [20]. The magnets attached to the device are essential to this design because they produce an attractive power that facilitates mobility. The MCL-TENG can effectively respond to a weak impact and may be used to evaluate speed parameters and identify cracks without requiring a complicated setup. Figure 2b illustrates a nanogenerator inspired by the flight of insects (DFIB-TENG). Following the derivation of the dynamic model of the DFIB-TENG through the application of the Euler–Lagrange equation, numerical analysis was used to investigate the dynamic response. In order to provide power for its health monitoring and sensor network, a DFIB-TENG can be used to collect low-frequency bridge vibration energy [107]. A self-powered vibration sensor system is depicted in Figure 2c. This system can identify vibration characteristics from a dual-mode triboelectric nanogenerator (AC/DC-TENG), which can produce either alternating current (AC) or direct current (DC) in distinct operation zones. More importantly, once the vibration amplitude exceeds the danger threshold, the AC rapidly changes to DC, activating an alarm system immediately so it can correctly anticipate the construction hazard. This approach can be a simple means to figuring out how healthy a building’s structure is [108].

### 3.2. Self-Powered Sensor Based on the Triboelectric Nanogenerators for Tunnels

Tunnels provide a quicker path by cutting through large rock. In order to reduce transportation congestion, several tunnels are constructed in towns. Tackling the darkness of tunnels for inspection of mechanical and structural elements necessitates a reliable lighting system. Because most tunnels are located outside of cities, providing lighting is a significant challenge. TENGs have been able to tackle this problem to a substantial extent through energy harvesting.

Figure 3a is an illustration of a wireless and self-powered traffic sensor, where a hybridized nanogenerator composed of triboelectric nanogenerators and electromagnetic generators serves as the primary source of sustainable power. It has been demonstrated that a hybridized nanogenerator can effectively catch wind energy generated by a vehicle that is driving through a tunnel. This study reveals new applications for triboelectric nanogenerators, such as high-performance energy harvesting from ambient mechanical sources and serving as long-term power sources for wireless traffic volume sensors [109]. Figure 3b reports a bionic TENG tree that is meant to harness wind energy by leveraging the coupling effects of contact electrification and electrostatic induction. The tree is made up of supercells, with cell I in the leaf and cell II in the stem, and it is made up of supercells. Because of its benefits of portability, low cost, simple production, and appealing appearance, the triboelectric nanogenerator tree has significant applications in a variety of fields, including the illumination of billboards, the measuring of vehicle speeds, and other sectors [110].

### 3.3. Self-Powered Sensor Based on the Triboelectric Nanogenerators for Highways

Highways are the most common types of roads that we see on a daily basis. They are designed to accommodate both automobiles and bicycles and allow for movement over relatively short distances. Intelligent streets are necessary for maintaining both comfort and safety. The cost of maintaining the intelligent road electronics that are currently in use is typically performed by humans, and power is typically supplied through cable. The self-powered highway sensor based on TENG is depicted in Figure 4 and Figure 5. The traffic warning sensor-based multimode triboelectric nanogenerator (M-TENG) is depicted in Figure 4a [111]. In order to power traffic warning systems and signal sensors the M-TENG makes use of the mechanical energy generated when vehicles travel on normal roadways or bridges. Wind or mechanical vibrations may be the source of this energy. The M-TENG is a great tool for gathering wind power. After five minutes of charging the energy management unit circuit, the M-TENG can transmit RF signals while replicating the natural world. Based on the numerous uses listed above, the M-TENG not only demonstrates its exceptional energy harvesting capabilities but also highlights its enormous potential in the field of mobility. A high-performance, dual-mode triboelectric nanogenerator (AC-TENG) and a direct current triboelectric nanogenerator (DC-TENG) are depicted in Figure 4b [112]. Both of these nanogenerators are able to gather wind energy in an effective manner while simultaneously monitoring wind speeds in real time.

When compared to previous attempts, the material optimization results in an increase in one factor in the charge density provided by the AC-TENG. In addition, the AC-TENG demonstrates an amazing lifespan by combining an elastic construction and material optimization to create a low-friction force. As a result, it retains 87% of its electric power after 1,200,000 running cycles, making it one of the most durable products. Figure 5a depicts a hybrid self-powered smart sensing network (NG). TENG-based renewable energy harvesting was developed in a previous study, as was the smart sensing network node, which scavenges ambient energy for multifunctional monitoring. This design can generate sustainable power by capturing wind and solar energy simultaneously or independently [61]. The air breakdown model of a noncontact TENG for maximum charge density is shown in Figure 5b together with a floating self-excited sliding TENG (FSS-TENG) with self-excited amplification between the rotator and stator. The study provides an excellent method for improving floating sliding TENG output and can be used to collect other micromechanical energy [113].

### 3.4. Smart Infrastructure for Harvesting Energy from Roads

The intelligent infrastructure that is employed to absorb energy from roadways is depicted in Figure 6 and Figure 7. Figure 6a shows a representation of an omnidirectional hybrid triboelectric–electromagnetic nanogenerator, also known as an OD-HNG. This particular variety of nanogenerators is designed to gather vibrational energy from all different directions. In order to make the most efficient use of the restricted area, triboelectric nanogenerators (TENGs) are built using a multilayered structure. Also, the electromagnetic generator (EMG) unit has been hybridized in order to achieve an even higher level of output from the OD-HNGs. The tower spring was built in a manner that allows the OD-HNG to respond to external stimulation in both the vertical and horizontal directions. Also, the working bandwidth of the OD-HNG is extended, which enables it to carry out omnidirectional and broadband vibration energy harvesting in a manner that is exceptionally effective. Furthermore, by gathering vibration energy from the region around the vehicle, the OD-HNG can continuously charge transportable electrical devices as well as environmental sensors. This energy can be harvested from the environment. According to these parameters, the tower-based, spring-based OD-HNG is capable of very efficient omnidirectional and broadband vibration energy harvesting and is an attractive option for large-scale vibration energy harvesting. This indicates that the OD-HNG can harvest vibrational energy in a variety of directions and across a broad frequency range [114]. Triboelectric nanogenerators provide the foundation for the self-powered over-speed wake-up alarm system (SOWAS) that is seen in Figure 6b. A SOWAS is made up of the following components: an energy harvesting triboelectric nanogenerator (also known as an E-TENG), an energy management module (also known as an EMM), an overspeed sensing triboelectric nanogenerator (also known as an S-TENG), and a power switch module (also known as a PSM). The SOWAS that was designed is capable of operating in unattended traffic situations. During this time, a full system was put in place to watch for aspects like going too fast and using too much energy [44].

Figure 7a provides a type of origami tessellation (OT) foundation that improves the electric output performance of TENG and permits its deployment in road pavement for energy harvesting. The OT foundation, thanks to the robustness of the structure, makes it possible to have multiple layers of facets for the installation of turbo pairs and can be controlled by very minute stimuli. It is possible to make effective use of the OT foundation in either traction or compression; however, this is dependent on the initial configuration of the OT foundation. In addition, the capacity of the OT base to fold up into itself makes it possible for the intended gadgets to readily fit into a space that is only a few centimeters wide. In this design, the devices will supply future intelligent transportation systems with clean energy that is both practical and usable. The findings of this research have led to the development of a ground-breaking OT-TENG design that has the potential to greatly increase the output performance of TENG devices in a more generic shape space and flexible environment [115]. Figure 7b presents a magnetic lifting triboelectric nanogenerator, also known as an ml-TENG, that can be utilized for active sensing and energy harvesting under cyclic loading conditions, such as those encountered in traffic. When operating in sliding mode, the ml-TENG makes use of magnetic force to generate a repulsion force that initiates a relative motion between the electrode and the dielectric layers. This motion is caused by the attraction and repulsion of these two forces, demonstrating that the ml-TENG is a potent device that can be used to build active sensing systems for real-world uses such as velocity detection [116].

### 3.5. Self-Powered Vehicle Sensors Based on Triboelectric Nanogenerators for Road Intelligent Systems

Figure 8 is a representation of the self-powered vehicle sensors that are utilized for road-based intelligent systems. These sensors are based on triboelectric nanogenerators and provide electricity themselves. Figure 8a illustrates one example of an integrated self-powered Hall vehicle sensor (SPHVS) for an automotive safety system. This sensor consists of a magnet, a Hall element, a triboelectric nanogenerator, and a management circuit, among other components. It is envisaged that the triboelectric source of continuous current will be able to provide power to the Hall element in rotation at any speed. In addition, the performance of the SPHVS was comprehensively investigated for the purposes of sensing vehicle speed and monitoring vehicle brakes. The results of this evaluation show that the SPHVS possesses a vast sensing range, high stability, and extraordinary sensitivity [117]. Guo et.al. (2018) demonstrated a workable method for speed detection and braking monitoring. An on-vehicle magnetically triboelectric nanogenerator (V-TENG) is depicted in Figure 8b. This on-vehicle nanogenerator captures the spinning energy of the tires to provide a direct power source for the tire pressure sensor. The V-TENG is a triboelectric generator that works on the principle of contact electrification and consists of two separate triboelectric materials with opposite polarities. The distinguishing advantages of the seesaw balance structure include the capacity to mitigate the effects of a strong centrifugal force even when the device is operating at high speeds and a quadrupling of the efficiency of the contact areas of a relatively compact device. The swinging movement of many V-TENGs that are attached to the wheel hub is occasionally activated by magnets that do not come into contact with the brake caliper. At the temperature that triggers the alarm, the V-TENG effectively creates electrical energy by using a thermally stable polymer sheet [118]. Moreover, it has been demonstrated that the V-TENG can provide power to a commercial wireless sensor, which can then send real-time temperature data to a receiver and display those data on a monitor interface. The bearing structural TENG (BS-TENG) is shown in Figure 8c, and it is based on a 3D-printed structure that has a diversified design. It is capable of reaching a rotating speed of more than 1500 rpm. These two qualities must be present for drivers to be able to operate their automobiles in a secure manner. Within the scope of this investigation, a strategy for increasing TENGs’ operational frequency was proposed. In the event that it is successful, this strategy may present brand new potential for the application of TENGs in the driving systems of intelligent vehicles [119].

### 3.6. Smart Pedestrian Crossing System Based on the Triboelectric Nanogenerators

Figure 9 illustrates an intelligent pedestrian crossing system utilizing triboelectric nanogenerators. Figure 9a presents and develops a novel type of fabric rebound triboelectric nanogenerator (FR-TENG) that may be utilized for effective energy harvesting and self-powered sensing. With the purpose of enhancing the electric performance of FR-TENGs, a comprehensive investigation of their structural properties was also conducted. The electrical performance of the FR-TENG in its as-manufactured state has been shown to be reliable in terms of energy harvesting, cycle washing capabilities, and mechanical durability. This system is constructed using FR-TENGs. In the study, a fresh perspective on textile-only TENGs was presented, and a sophisticated human–machine software interface for sensing applications was demonstrated [120]. Figure 9b shows how a cement-based TENG for large-scale energy harvesting from human footfall might be improved. This can be accomplished by modifying the material composition of the TENG. Combining CB nanoparticles with hydroxyethyl cellulose (HEC) results in the production of a cement–carbon black (CB) composite, which can be utilized as a triboelectric material for TENGs. This is so that a major improvement may be achieved, which requires increasing the triboelectric charge density. This is the most important factor in deciding whether or not the dielectric constant is large. This is because the HEC admixture serves as a trigger for both processes. This novel method has the potential to clear the way for the development of intelligent cement materials, which could then be used to implement uses in large-scale energy harvesting, paving the way for the development of sustainable, clean, and green energy sources [121]. Figure 9c shows a suggested TENG with high energy conversion efficiency and an impact output that lasts a long time. This is a design for a noncontact, freely oscillating TENG that utilizes a freely oscillating structure without frictional contact and demonstrates sustained output after impacts. It has the ability to illuminate 20 LEDs for 10 s with a single, instantaneous impact. After twelve collisions, this platform was used to construct a self-cleaning solar panel system that efficiently cleaned 79.2% of the dust from the test panel surface [122]. Ma et.al. (2022) investigated the mechanism for gathering energy from impacts and a practical application for solar panels that clean themselves. Figure 9d shows how the paint-based TENG, also called the PBT, is made using a simple spray deposition method to collect the lost mechanical energy released by the paint-coated surface when it comes into contact. The electrical outputs of the PBT were studied, and a maximum output increase of 280 percent was reported. In addition, the versatility of the PBT was proven by its manufacturing using a variety of materials, and the PBT has excellent stability. When all of the applications have been processed, the intrusion detection system (IDS) and the camouflaged keypad (CKP) are ultimately installed. The CKP effectively substitutes for the standard keyboard’s keypad, ensuring a high level of personal security throughout the login procedure [123].

### 3.7. Self-Powered Sensors Based on Triboelectric Nanogenerators for Monitoring Driving Behaviors

Figure 10 and Figure 11 depict self-powered sensors for monitoring driving behaviors that are based on TENG. In Figure 10a, intelligent systems based on triboelectric nanogenerators are proposed to provide real-time driver status monitoring and fatigue warnings. The intelligent system is made up of a signal processing unit and self-powered steering-wheel angle sensor (SSAS) [124].

The SSAS, which consists of a stator, a rotor, and a sleeve, is mounted on the steering shaft. To improve the sensor’s precision, the electrodes have a phase difference. The SSAS is responsible for recording the angle at which the driver turns the steering wheel. In the meantime, the signal processing unit is responsible for determining a warning threshold for each parameter by analyzing the recorded data from the driver, including the number of revolutions, the average angle, and any other relevant data.

By comparing these metrics and threshold levels, the system evaluates the state of the driver in real time. This technique has significant potential implications in the domains of intelligent driving and traffic safety. Figure 10b depicts the development of a self-powered smart safety belt that monitors the turning motions of the driver as well as the forward position of the vehicle. This design is made up of two different types of triboelectric nanogenerator (TENG) [106]. AS-TENGs are produced by incorporating two TENGs, locating them on the top and bottom arches so that there is continuous and gradually growing contact between the triboelectric layers in both arches. This enables the AS-TENG to work properly. This research enhances the spectrum of TENGs uses in wearable electronics by providing an efficient method for incorporating self-powered TENGs sensors into an existing driving environment for driving-state monitoring. A novel sweep-type triboelectric nanogenerator (ST-TENG) is depicted in Figure 11a, where it is shown to be constructed out of a push rod, shells, two flywheels, and one freewheel. An ST-TENG has the ability to collect energy from arbitrary motion triggers and observe the behavior of drivers. It has been found that the data collected by the ST-TENG can be used to represent road conditions and the behavior of drivers. In addition, there is a possibility that ST-TENGs will contribute to the growing acceptance of intelligent driving systems [125]. Figure 11b shows a sector-shaped triboelectric nanogenerator (TENG) for measuring the accelerator and brake pedals and a DT-TENG for sensing a vehicle’s steering angle. Because of its extraordinary sensitivity, the DT-TENG needs only two channels in order to successfully record signals. The TENG categorizes the driver’s driving behavior as either safe or aggressive based on the collected data, and the results are sent back to the driver so that they can assess and modify their own driving habits appropriately. This demonstrates that a self-powered, high-sensitivity TENG-based sensor can be built and used in autonomous transit systems [79].

## 4. Materials for Triboelectric Nanogenerator-Based Self-Powered Sensors in Transportation Infrastructure including Bridges, Highways, and Tunnels

Self-powered sensors based on triboelectric nanogenerators (TENGs) offer promising opportunities for transportation applications, including bridges, tunnels, highways, smart infrastructure, road intelligent systems, smart pedestrian crossing systems, and monitoring driving behaviors. The selection of appropriate materials is crucial for optimizing the performance and efficiency of TENG-based sensors in each application. Here, we discuss the different materials commonly employed in these specific scenarios:Self-powered sensor based on triboelectric nanogenerators for bridges: Polymeric materials: polydimethylsiloxane (PDMS), polytetrafluoroethylene (PTFE), polyvinylidene fluoride (PVDF), and their composites. Conductive materials: indium tin oxide (ITO), graphene, carbon nanotubes (CNTs), and silver nanowires. Substrate materials: flexible and durable materials such as polyimide, polyethylene terephthalate (PET), or polyethylene naphthalate (PEN). Encapsulation materials: to protect the TENG-based sensor from environmental factors, materials like epoxy resins or silicone elastomers are commonly used.Self-powered sensor based on triboelectric nanogenerators for tunnels: Similar to bridge applications, polymeric materials (PDMS, PTFE, PVDF) and conductive materials (ITO, graphene, CNTs) are frequently used. Special attention is given to materials with high durability and resistance to moisture, dust, and harsh underground conditions.Self-powered sensor based on triboelectric nanogenerators for highways: Materials used in highway applications are typically selected for their robustness, flexibility, and resistance to weathering. Flexible polymers, such as PDMS, PTFE, and PVDF, are commonly employed due to their resilience and ability to withstand external forces. Conductive materials, including ITO, graphene, CNTs, and silver nanowires, are used to ensure efficient charge transfer and conductivity.Smart infrastructure to harvest energy from roads: Materials used for energy harvesting systems integrated into roads should possess high piezoelectric or triboelectric properties. Piezoelectric materials: lead zirconate titanate (PZT), zinc oxide (ZnO), polyvinylidene fluoride (PVDF), and its copolymers. Triboelectric materials: PDMS, PTFE, PVDF, and their composites, as well as materials with high triboelectric coefficients, like polyamide or polyethylene.Self-powered vehicle sensors based on triboelectric nanogenerators for road intelligent systems: Similar to bridge and highway applications, flexible polymers (PDMS, PTFE, PVDF) and conductive materials (ITO, graphene, CNTs) are commonly used. Additionally, pressure-sensitive materials, such as pressure-sensitive films or elastomers, are employed to capture vehicle-related parameters.Smart pedestrian crossing system based on triboelectric nanogenerators: Materials used in smart pedestrian crossing systems require good triboelectric properties and mechanical resilience. PDMS, PTFE, PVDF, and their composites are commonly used for the triboelectric layers. For the structural components, flexible and durable materials like PET, PEN, or elastomers are employed.

## 5. Challenges, Perspective, and Insight for Self-Powered Sensors for Intelligent Road Environments

In addition to intelligent automobiles, intelligent roads are also a significant component. Figure 12 shows problems, perspectives, and insights for self-powered sensors for intelligent road environments. There are several technical weaknesses and issues with TENGs [126,127,128,129]. The most significant of these are fatigue, the possibility of losing elasticity, the possibility of damaging auxiliary parts (like lever arms and springs), and the difficulty of absorbing electrical energy from oscillations with low frequency and low amplitude. One of the most challenging tasks is ensuring that a TENG can withstand repetitive loads [130]. TENGs are also influenced by coupling factors including temperature, humidity, and pressure, as well as extra components such as UV radiation and the like. High traffic loads and severe environmental factors have the potential to make a TENG unstable, resulting in erroneous monitoring and irregular operation [131]. A feasible solution can be found in the design of a fully packed structure; however, the issue of whether or not the packaged structure is compatible with the road structure needs to be answered. Incompatibility not only reduces the electrical performance of TENGs, but it also greatly reduces the amount of time they can be used for road operation. As a result, ensuring the consistency of TENGs remains a difficult task.

Finally, there are several materials, designs, and structures that hold promise for further exploration and development.

Materials: Researchers should continue to investigate advanced materials with enhanced triboelectric properties and mechanical resilience. This includes exploring new polymeric materials, such as polymethylpentene (PMP), thermoplastic polyurethane (TPU), or composites with improved performance. Additionally, the exploration of novel conductive materials like carbon-based nanomaterials, metal oxides, or conducting polymers can contribute to higher charge transfer efficiency.Designs: The design of TENGs for intelligent transportation infrastructure should focus on maximizing energy conversion efficiency and system integration. Novel design approaches, such as multilayered structures, hierarchical structures, or patterned surface modifications, can enhance the overall performance of TENGs. Exploring innovative electrode designs, such as microstructured surfaces or three-dimensional architectures, can also improve energy harvesting capabilities.Structures: Researchers should explore adaptable and scalable TENG structures to accommodate various transportation infrastructure applications. Developing flexible and conformable TENGs that can be easily integrated into different surfaces, such as bridges, highways, and tunnels, is crucial. Additionally, investigating self-powered sensing systems with distributed TENGs to enable large-scale monitoring and data collection would be beneficial.

## 6. Conclusions

According to self-powered sensors based on triboelectric nanogenerators (TENGs) with attractive characteristics of high sensitivity, low limitation, and high efficiency, TENGs have recently been researched and have significantly benefited self-powered sensing. The TENG is anticipated to contribute significantly to the construction of smart roads, which will benefit the entire road system, including bridges, tunnels, and freeways. A number of the applications, including energy harvesting, self-powered sensing, and self-adaptation, have eventually been developed and come into use as the process has continued. The information in this article will be useful for industrialists associated with transportation engineering, including bridges, highways, and tunnels who work on self-powered sensors, energy harvesting, and sustainability. In the present study, the application of self-powered sensors based on triboelectric nanogenerators (TENGs) in intelligent road environments was described along with the monitoring of driving behavior. An overview of triboelectric nanogenerators for intelligent road environments was also presented. Table 2 presents the performance output comparison of various TENG devices for both monitoring driving behavior and self-powered sensors influenced by triboelectric nanogenerators (TENGs) such as self-powered sensors for bridges, tunnels, highways, road intelligent systems, monitoring driving behaviors, smart pedestrian crossing systems, and smart infrastructure to harvest energy from roads. Most of the electrodes used in TENGs are Al and Cu and also most of the triboelectric layers are PDMS, PTFE, and Kapton.

## Figures and Tables

**Figure 1 sensors-23-06634-f001:**
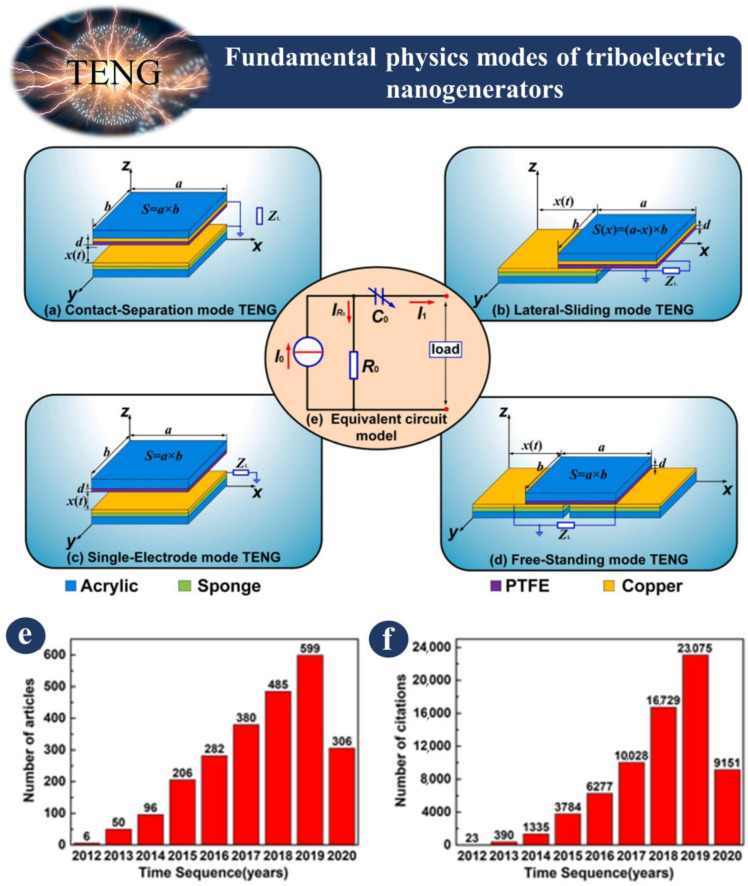
Triboelectric nanogenerator modes: (**a**–**d**) contact separation mode, lateral sliding mode, single electrode mode, and free standing triboelectric layer mode [78]. (**e**) The number of TENG research articles published each year. (**f**) The number of citations of TENG research articles each year [79].

**Figure 2 sensors-23-06634-f002:**
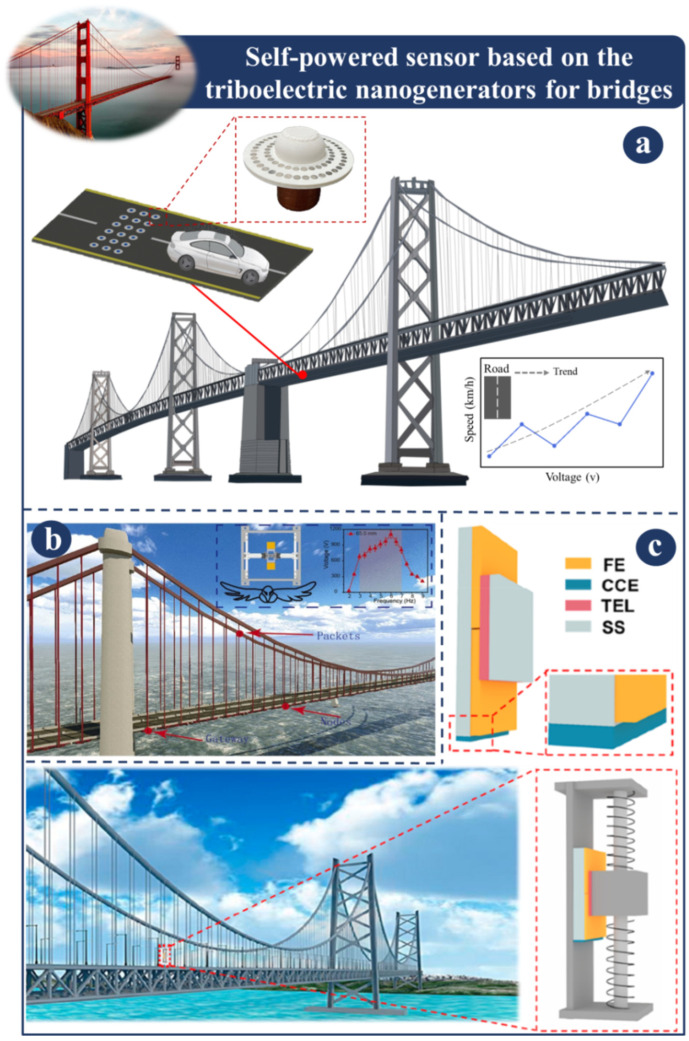
Demonstration of self-powered sensor based on a TENG for bridges: (**a**) Application of MCL-TENG as self-powered sensor [20]. (**b**) The design principle of DFIB-TENG [107]. (**c**) An AC/DC-TENG’s operational mechanism [108].

**Figure 3 sensors-23-06634-f003:**
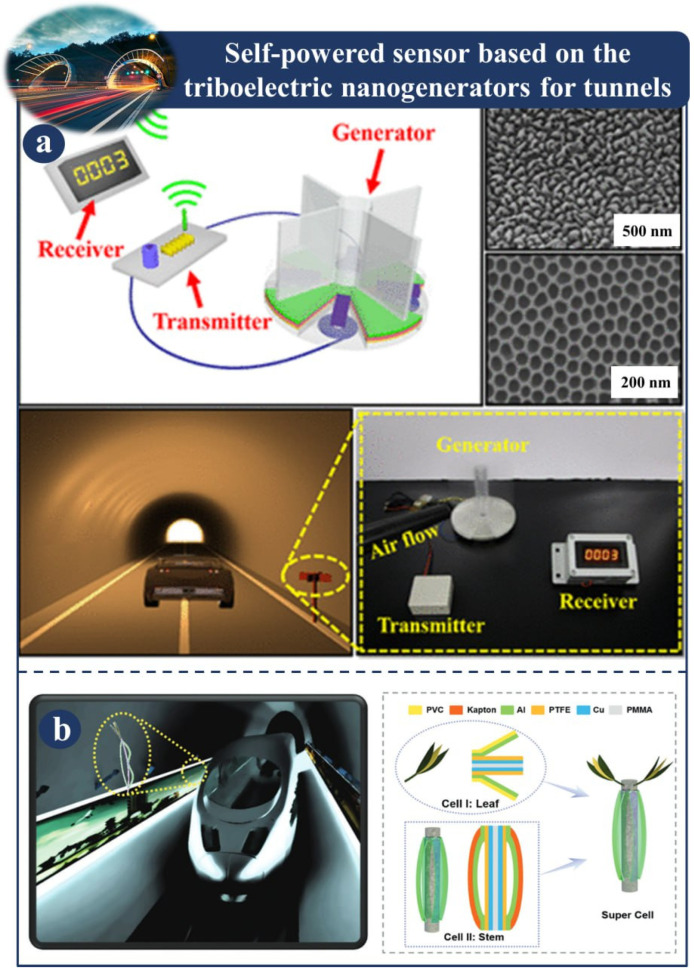
Demonstration of self-powered sensors based on a TENG for tunnels: (**a**) Design of a framework for a wireless system that is self-powered and measures traffic volume [109]. (**b**) Fabrication and design of a flexible TENG tree [110].

**Figure 4 sensors-23-06634-f004:**
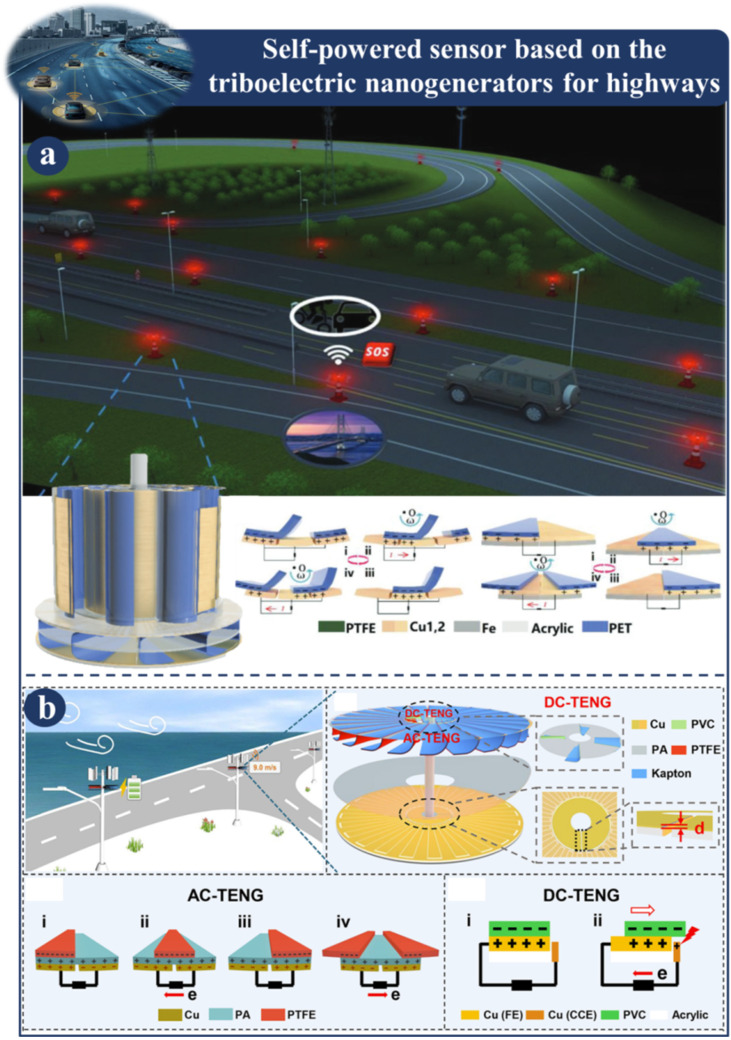
Demonstration of self-powered sensor based on a TENG for highways: (**a**) Fabrication and structural design of M-TENG [111]. (**b**) Fabrication and various scenarios of the dual-mode TENG [112].

**Figure 5 sensors-23-06634-f005:**
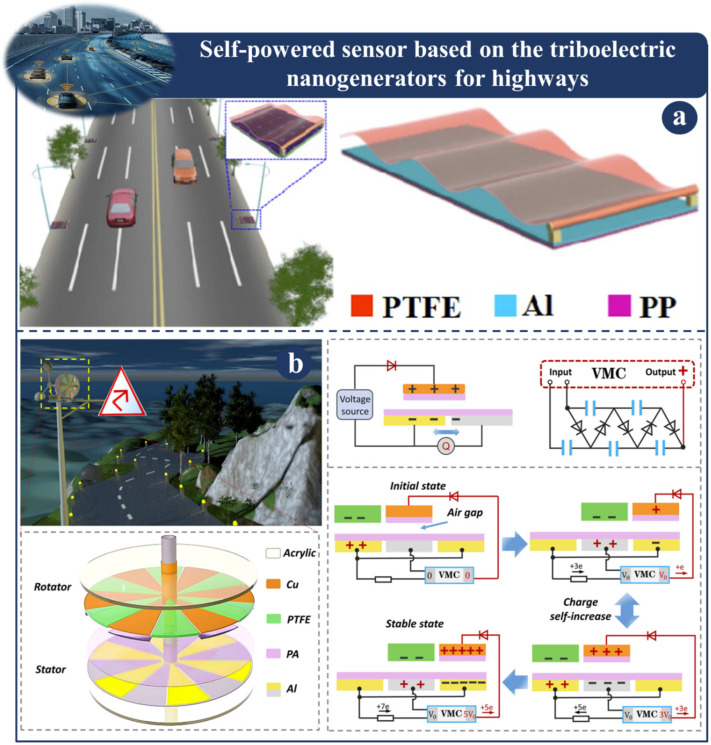
Demonstration of self-powered sensor based on a TENG for highways: (**a**) Intelligent traffic control system hybrid NG illustration [61]. (**b**) FSS-TENG structure and design principles [113].

**Figure 6 sensors-23-06634-f006:**
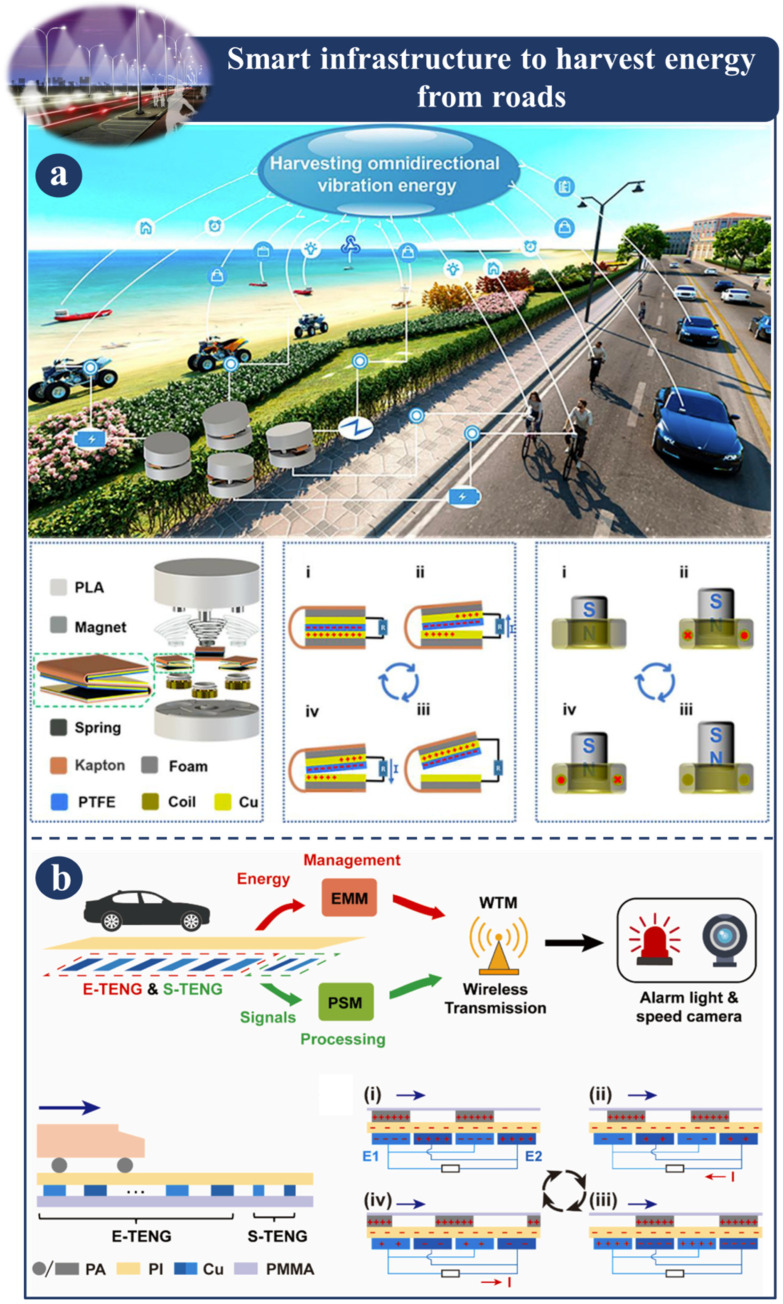
Demonstration of smart infrastructure to harvest energy from roads: (**a**) Fabrication, structural design, and application of OD-HNG [114]. (**b**) Overspeed wake-up alarm system powered by triboelectric nanogenerators (SOWAS) [44].

**Figure 7 sensors-23-06634-f007:**
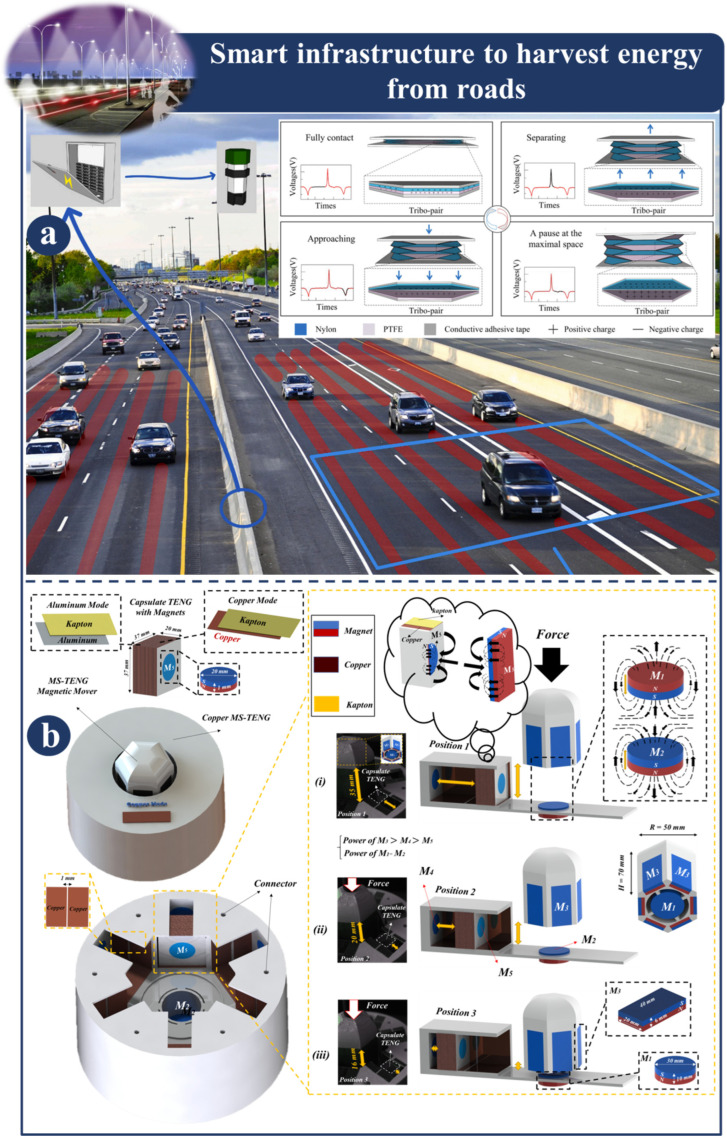
Demonstration of smart infrastructure to harvest energy from roads: (**a**) Application and design principal of OT-TENG [115]. (**b**) Structure and various scenarios of ml-TENG [116].

**Figure 8 sensors-23-06634-f008:**
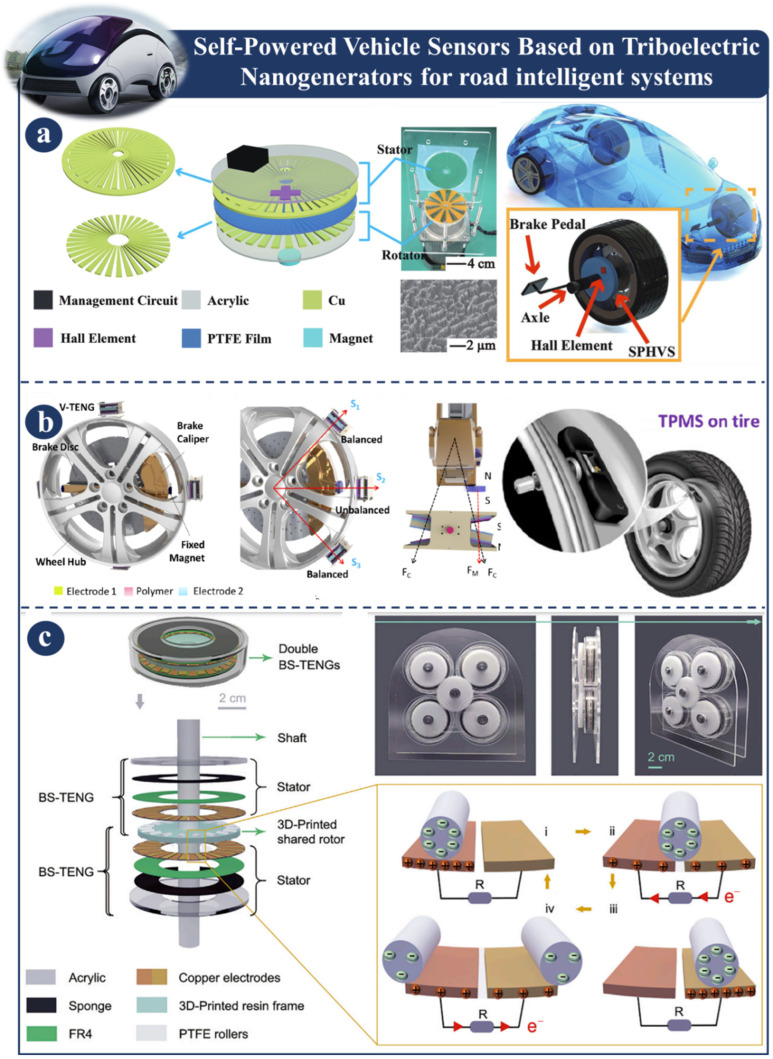
Demonstration of self-powered vehicle sensors based on TENG for road intelligent systems: (**a**) Design principles of SPHVS sensor [117]. (**b**) Fabrication and design principal of V-TENG [118]. (**c**) The structural design and functioning mechanism of a BS-TENG [119].

**Figure 9 sensors-23-06634-f009:**
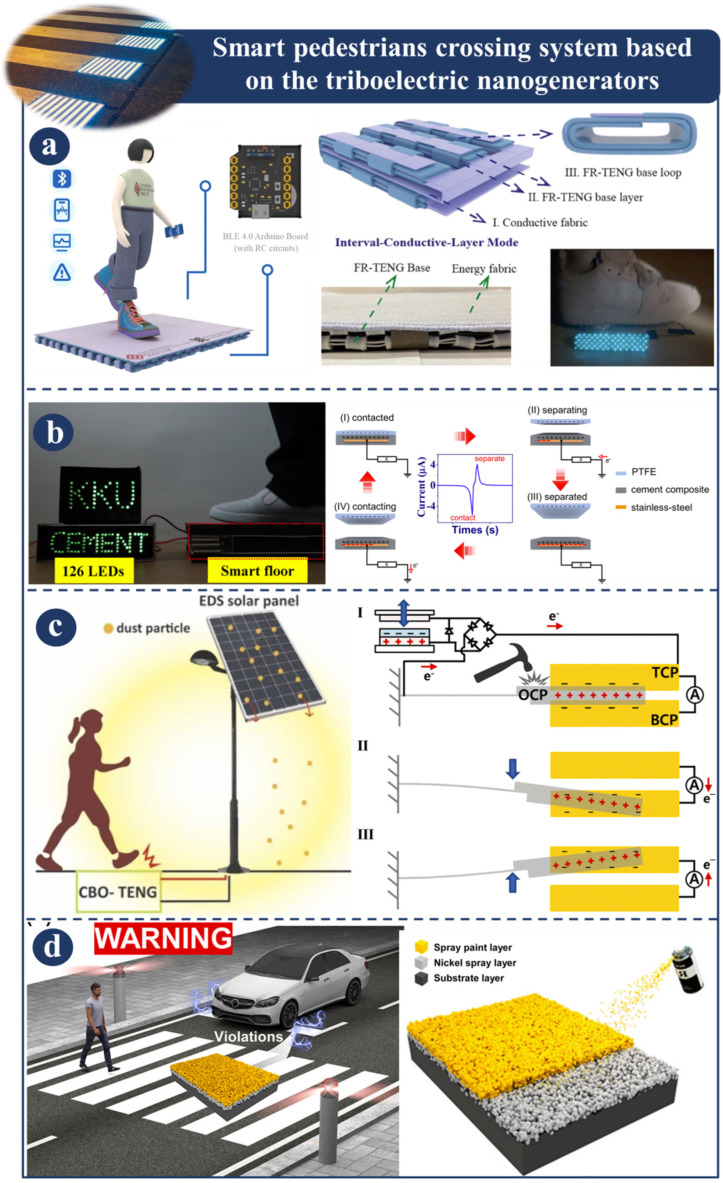
Demonstration of smart pedestrian crossing system based on triboelectric nanogenerators: (**a**) Application, principal, and fabrication of FR-TENGs [120]. (**b**) The working mechanism of cement-based TENG [121]. (**c**) Design principle and application of CBO-TENG [122]. (**d**) PBT’s illustration and mechanism of operation [123].

**Figure 10 sensors-23-06634-f010:**
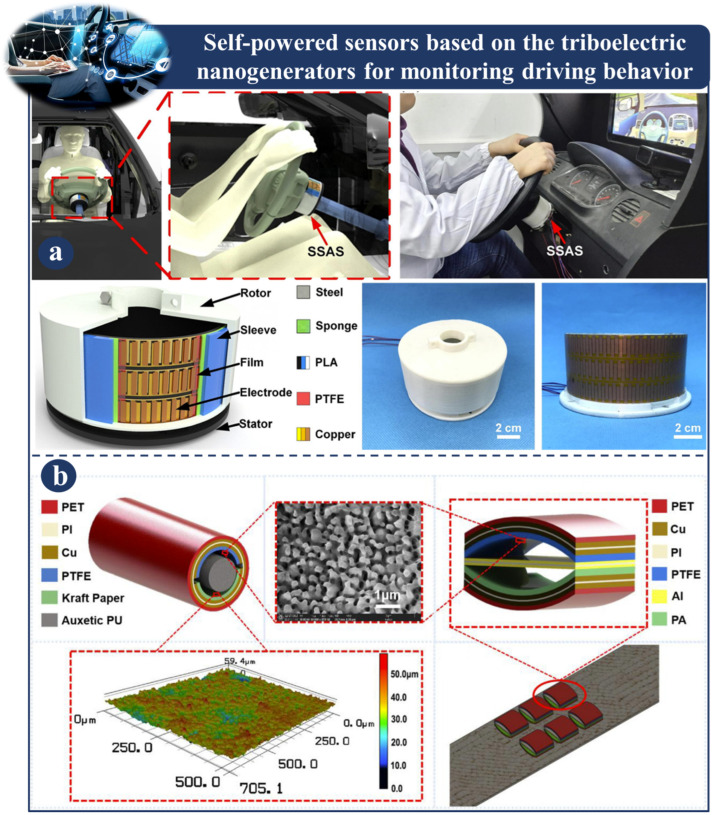
Demonstration of self-powered sensors based on a TENG for monitoring driving behaviors: (**a**) Application and design principal of (SSAS) [124]. (**b**) The design principle of APU-TENG and AS-TENG [106].

**Figure 11 sensors-23-06634-f011:**
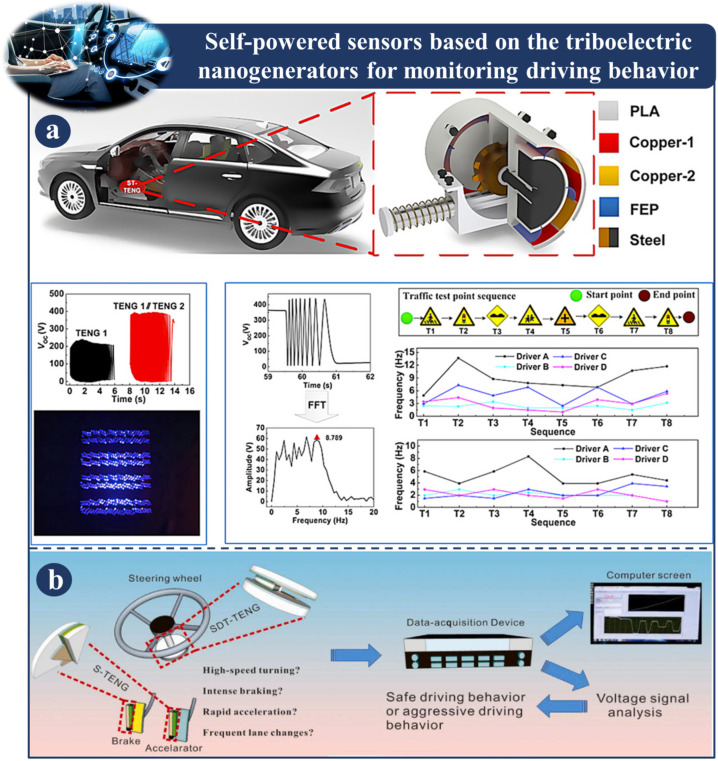
Demonstration of self-powered sensors based on a TENG for monitoring driving behaviors: (**a**) Application and design principal of an ST-TENG [125]. (**b**) Fabrication and application of DT-TENG [79].

**Figure 12 sensors-23-06634-f012:**
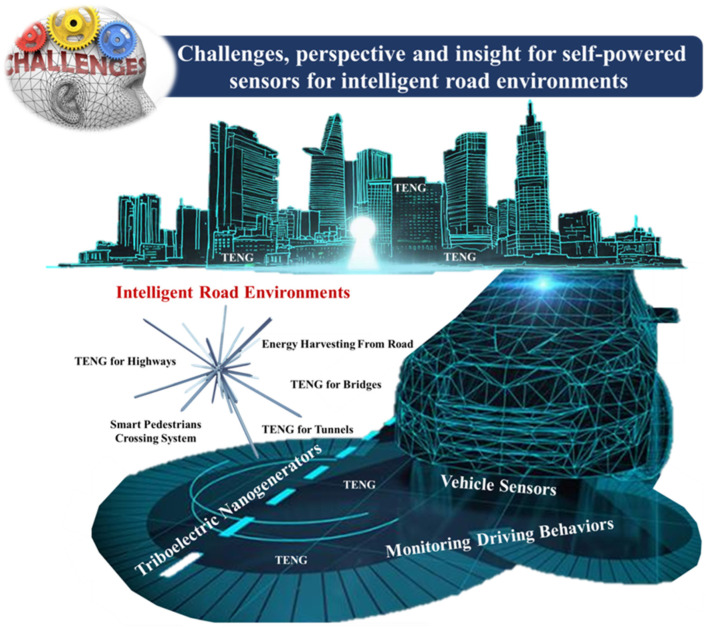
Demonstration of Challenges, perspective, and insight for self-powered sensors for intelligent road environments.

**Table 1 sensors-23-06634-t001:** Summary of the different ways of vehicle energy harvesting.

Method	Description
Kinetic energy harvesting	This approach involves harnessing the vehicle’s kinetic energy during braking or deceleration using regenerative braking systems, which convert mechanical energy into electrical energy. Triboelectric nanogenerators can also capture energy from friction between two materials in contact, such as tires and the road surface.
Vibration energy harvesting	Vibrations generated by vehicle movement or engine operation can be converted into electrical energy. Piezoelectric nanogenerators utilize piezoelectric materials that produce an electric charge in response to mechanical stress, enabling the conversion of vehicle vibrations into usable power.
Solar energy harvesting	Vehicles can incorporate solar panels to capture sunlight and convert it into electrical energy. Solar powered vehicles or solar charging systems integrated into the vehicle’s body or roof can harness the sun’s energy for auxiliary systems or charging the vehicle’s battery.
Thermoelectric energy harvesting	This method utilizes the temperature difference between the vehicle’s engine and the surrounding environment. Thermoelectric materials, including those based on triboelectric or piezoelectric principles, can convert this temperature gradient into electrical energy.
Wind energy harvesting	Moving vehicles create airflow, which can be utilized through small wind turbines or wind capturing structures. These devices convert the kinetic energy of the moving air into electrical power, contributing to vehicle energy harvesting.
Electromagnetic induction	By utilizing magnetic fields, electromagnetic induction harvests energy from the vehicle’s mechanical movement or changing magnetic fields. This method can involve road-embedded coils or electromagnetic systems that interact with the vehicle’s components, converting mechanical energy into electrical power.
Triboelectric nanogenerators	Triboelectric nanogenerators convert mechanical energy from vehicle motion or vibrations into electricity through the triboelectric effect. This effect is based on the charge transfer between two materials with different electron affinities when they come into contact or separate.
Piezoelectric nanogenerators	Piezoelectric nanogenerators utilize materials that generate electric charge in response to mechanical strain or vibrations. When subjected to vehicle vibrations or deformations, these materials can produce electrical energy.

**Table 2 sensors-23-06634-t002:** Summary of various TENG techniques for transportation infrastructure.

Structure	Year	Authors	Applications	Max Open- Circuit Voltage (V)	Max Short- Circuit Current	Surface Power Density/Power	TENG Mode
Self-powered sensors for bridges	2022	Jiao et al. [20]	Velocity sensing and damage detection	5.1 V	-	9 µW	FS
	2022	Luo et al. [107]	Harvesting low-frequency vibrations	1108 V	12.17 μA	1.25 mW	FS
	2020	Li et al. [108]	Vibration monitoring systems	30 V	200 nA	-	FS
Self-powered sensors for tunnels	2016	Zhang et al. [109]	Traffic volume sensors	3.5 V	5 mA	17.5 mW	LS
	2018	Bian et al. [110]	Harvesting wind energy	330 V	59.6 µA	3.6 mW	CS
Self-powered sensors for highways	2022	Yang et al. [111]	Traffic monitoring and warning	968.82 V	14.103 µA	22 W	FS
	2022	He et al. [112]	Windspeed monitoring	1140 V	0.64 mA	267.3 mW	FS/LS
	2017	Wang et al. [61]	Multifunctional sensing	140 V	1 μA	-	CS
	2021	Long et al. [113]	Energy harvesting	150 V	17.6 μA	16.7 mW	FS
Smart infrastructure to harvest energy from roads	2022	Cao et al. [114]	Vibration energy	1.75 V	-	-	CS
	2022	Cao et al. [44]	Intelligent transportation	150 V	1.3 μA	42 mW	FS
	2020	Zhang et al. [115]	Energy harvesting	-	-	20.25 µW	CS
	2021	Matin Nazar et al. [116]	Energy harvesting and active sensing	4 V	-	-	FS
Self-powered vehicle sensors for road intelligent systems	2018	Guo et al. [117]	Vehicle sensors	140 V	25 µA	4.2 mW	FS
	2018	Qian et al. [118]	Tire pressure monitoring system	316 V	-	22.3 mW	CS
	2021	Yang et al. [119]	Intelligent vehicle monitoring	72 V	76 µA	-	FS
Smart pedestrians crossing system	2022	Jiang et al. [120]	Monitoring of human motions	418.09 V	65.85 µA	199.14 µW·cm^−2^	CS
	2022	Kuntharin et al. [121]	Energy harvesting	-	-	2.38 W/m^2^	SE
	2022	Ma et al. [122]	Daily life self-cleaning solar panel	1300 V	16 µA	-	CS
	2021	Yun et al. [123]	Smart traffic system	80 V	7.2 µA	-	SE
Self-powered sensors for monitoring driving behaviors	2021	Xu et al. [124]	Monitoring driver	9 V	-	-	FS
	2019	Feng et al. [106]	Monitoring driver	6.98 V	24.53 nA	-	CS
	2020	Xie et al. [125]	Energy harvesting/monitoring driver	400 V	15 µA	-	FS
	2021	Lu et al. [79]	Monitoring driver	220 V	-	-	SE

## Data Availability

The data presented in this study are available upon request from the corresponding author.

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
