# Peer review of "Advancements in Triboelectric Nanogenerators (TENGs) for Intelligent Transportation Infrastructure: Enhancing Bridges, Highways, and Tunnels"

_sensors, 2023, doi:10.3390/s23146634_

Round 1
Reviewer 1 Report
This manuscript discusses both monitoring driving behavior and self-powered sensors influenced by triboelectric nanogenerators (TENG). It also considers energy harvesting and sustainability in smart road environments such as bridges, tunnels, and highways. Furthermore, the information gathered in this study can help the readers enhance their knowledge concerning the advantages of employing these instruments for innovative uses of their powers.
The reviewer recommends the paper for publication after minor revisions. The following comments listed below are expected to assist with further improving the quality of the manuscript:
1. The reviewer suggests that the author add a diagram in Fig.1 to describe several typical materials, structures, and mechanisms used for vehicle road energy harvesting.
2. The reviewer suggests that the author add relevant visualized images to demonstrate the future development trends of TENG in Bridges Highways and Tunnels.
3. Energy harvesting is an innovative and promising method for Intelligent Transportation Infrastructure, the reviewer suggests the author to add some work in the introduction to further improve the breadth of article research. Nano Energy, 2023: 108222. Applied Energy 314 (2022): 118983.
4. Add more comparative tables to summarize the different ways of vehicle energy harvesting.
5. The paper is well-written, but not free from grammatical errors. Please check the whole manuscript for grammatical errors.
good
Author Response
Dear reviewer, please find the attachment.

Reviewer 2 Report
Authors have reported studies on improvements to the efficiency, efficacy, and sensitivity of self-powered sensing Triboelectric Nano generators. Paper seem like a report rather than a review paper. Previous work should be presented in quantitative form not in the qualitative approach. No prior studies have been compared. Commentaries of the earlier works have been written. Therefore, I must reject this work and should not be evaluated again until or until it do not improve considerably.
Moderate editing of English language required
Author Response

(The authors gave the same response as above.)

Reviewer 3 Report
The authors reported a very interesting review of TENGs. The design of the manuscript is good and I support publishing it after some minor corrections as the following:
- The manuscript needed more literature review about the different materials that can be used in each of these applications.
- Can you add also a summary of the potential materials for this application in the introduction part
- By the end of section 4, please add which materials, designs, and structures would you suggest for the readers to keep working on this topic
- Many typing errors should be considered
- Figure organization: While the figures provided in the manuscript are informative, I believe it would be beneficial to present the figures of the application and the literature review separately. This separation would facilitate better tracking and comprehension for readers.
- Statistical figure: To provide a comprehensive overview of the field, I recommend including a statistical figure at the beginning of the manuscript. This figure could present the number of papers published in this particular field up until the present time. Such data would greatly enhance the context and significance of the research presented.
- Table 1 placement: It would be advantageous to relocate Table 1 to the beginning of the manuscript. By positioning it early on, readers will have immediate access to the relevant information and findings, enabling them to better grasp the research content.
Many typing errors should be considered
Author Response

(The authors gave the same response as above.)

Reviewer 4 Report
The present review discusses both monitoring driving behavior and self-powered sensors influenced by triboelectric nanogenerators (TENG). The fundamental physical modes of triboelectric nanogenerators, self-powered sensors for use in intelligent road environments and the challenges, viewpoints, and thoughts related to the creation of self-powered sensors based on TENG for use in bridges, tunnels, and roads have been addressed. Some comments and suggestions are given as below.
- The “TENG effect” in line 36 should be replaced by “triboelectric effect”.
- “Electrostatic charges” in line 72 should be replaced by “Electrostatic induction charges”.
- What does the author mean by saying that “Every time a TENG layer is changed, a potential differential is produced because the state changes from electrostatic to non-electrostatic.” in lines 75 and 76.
- The language also needs to be polished, especially for the Abstract and Introduction sections.
The language needs to be significantly polished.
Author Response
Dear reviewer, please find attachment.

Reviewer 5 Report
The authors have described the sensors based on triboelectric nanogenerators (TENG) in intelligent road environments along with the monitoring of driving behaviour .
The figures are very informative and clear. Nice presentation of results!
The article is well –written however I suggest the authors to add more citations in the introduction section in order to extend the section in new energy harvester system
In this case I suggest to cite the following article:
Bijak, J., Lo Sciuto, G., Kowalik, Z., Trawiński, T., & Szczygieł, M. (2023). A 2-DoF kinematic chain analysis of a magnetic spring excited by vibration generator based on a neural network design for energy harvesting applications. Inventions, 8(1), 34.
Author Response

(The authors gave the same response as above.)

Round 2
Reviewer 3 Report
Thank you for your report
Moderate editing of English language required